# Canonical and Non-Canonical Wnt Signaling Generates Molecular and Cellular Asymmetries to Establish Embryonic Axes

**DOI:** 10.3390/jdb12030020

**Published:** 2024-08-02

**Authors:** De-Li Shi

**Affiliations:** 1Department of Medical Research, Affiliated Hospital of Guangdong Medical University, Zhanjiang 524001, China; de-li.shi@upmc.fr; 2Laboratory of Developmental Biology, Centre National de la Recherche Scientifique (CNRS), UMR7622, Institut de Biologie Paris-Seine (IBPS), Sorbonne University, 75005 Paris, France

**Keywords:** Wnt signaling, axis formation, dorsoventral axis, anteroposterior axis, left–right axis, Spemann organizer, left–right organizer, asymmetry formation, laterality defects

## Abstract

The formation of embryonic axes is a critical step during animal development, which contributes to establishing the basic body plan in each particular organism. Wnt signaling pathways play pivotal roles in this fundamental process. Canonical Wnt signaling that is dependent on β-catenin regulates the patterning of dorsoventral, anteroposterior, and left–right axes. Non-canonical Wnt signaling that is independent of β-catenin modulates cytoskeletal organization to coordinate cell polarity changes and asymmetric cell movements. It is now well documented that components of these Wnt pathways biochemically and functionally interact to mediate cell–cell communications and instruct cellular polarization in breaking the embryonic symmetry. The dysfunction of Wnt signaling disrupts embryonic axis specification and proper tissue morphogenesis, and mutations of Wnt pathway genes are associated with birth defects in humans. This review discusses the regulatory roles of Wnt pathway components in embryonic axis formation by focusing on vertebrate models. It highlights current progress in decoding conserved mechanisms underlying the establishment of asymmetry along the three primary body axes. By providing an in-depth analysis of canonical and non-canonical pathways in regulating cell fates and cellular behaviors, this work offers insights into the intricate processes that contribute to setting up the basic body plan in vertebrate embryos.

## 1. Introduction

Wnt signaling is critically involved in a wide variety of physiological and pathological processes, such as embryonic axis formation, cell proliferation, differentiation, polarity and migration, maintenance of cell or tissue homeostasis, congenital diseases, and cancer development [1,2,3]. Wnt pathways are evolutionarily conserved and can be divided into canonical and non-canonical branches, which elicit distinct biological responses in the cell. Canonical Wnt or Wnt/β-catenin signaling is dependent on the activity of β-catenin to induce target gene transcription and cell fate specification, while the two non-canonical Wnt branches, Wnt/planar cell polarity (PCP) and Wnt/Ca^2+^, are independent of β-catenin and act essentially as key regulators of cytoskeletal organization and cellular polarization [4]. Dysfunction of canonical and non-canonical pathways affects embryonic development and is closely associated with human disease. Extensive studies using vertebrate and invertebrate models have firmly demonstrated the essential role of these conserved signaling pathways in initiating the molecular and cellular asymmetries within the early embryo, which are indispensable for promoting the subsequent development of the dorsoventral (D-V), anteroposterior (A-P), and left–right (L–R) axes.

The formation of embryonic axes is a fundamental developmental process that helps to establish the basic body plan in each particular organism. It breaks the radial and bilateral symmetry of the embryo to promote subsequent organ morphogenesis. In many species, such as *Xenopus* and zebrafish, the D-V axis is specified by maternal determinants that are translocated to the dorsal region of the embryos after fertilization. The formation of the D-V axis also prefigures the A-P axis, which will be further patterned and elongated during gastrulation by combinatorial signaling and extensive morphogenetic movements [5,6]. Therefore, D-V patterning is tightly linked to the regionalization of the embryo along the A-P axis. Defects in D-V axis specification lead to the development of the embryo into a “belly piece”, without dorsal axial organs and consequently lacking the A-P axis [7]. The L–R axis is determined at relatively late stages of development in many vertebrates, generally at the end of gastrulation or during the stages of somitogenesis. This breaks the bilateral symmetry of the early embryo to influence the asymmetric morphogenesis of visceral organs, such as the rightward looping of the heart tube, and their appropriate positioning in the body cavity [8]. Disrupted L–R axis establishment is responsible for a spectrum of laterality defects, particularly congenital heart malformations [9]. It is well established that Wnt/β-catenin and Wnt/PCP pathways play critical roles in the formation of all three embryonic axes. Maternal Wnt/β-catenin signaling promotes dorsal axis specification while zygotic Wnt/β-catenin signaling regulates posterior development [5,10]. Non-canonical Wnt signaling, specifically the Wnt/PCP pathway, functions to elongate the A-P axis by coordinating cell movements and to initiate the L–R asymmetry by restricting cilia orientation in the L–R organizer [8,11]. Importantly, the participation of Wnt signaling in establishing the primary body axis is largely conserved in the Metazoa [10,12,13]. Molecular studies on the development of non-bilaterian animals such as hydras, sponges, and annelids suggest the existence of an ancient Wnt signaling center in the formation of body asymmetry [14,15]. However, the regulatory mechanisms underlying Wnt signaling in the fundamental developmental process of embryonic axis formation merit further investigations.

This review discusses the contribution of Wnt signaling to the establishment of asymmetry in vertebrate models. It highlights the current understanding of conserved regulatory mechanisms underlying the formation of D-V, A-P, and L–R axes. By providing an in-depth analysis of the interplay between Wnt/β-catenin and Wnt/PCP pathways in regulating cell fates and cellular behaviors, this work offers insights into the intricate processes that generate the basic body plan in vertebrate embryos.

## 2. Wnt Signaling Pathways

The developmental role of *Wnt* genes was first discovered in *Drosophila* by the identification of *wingles* (*wg*) mutants that showed disrupted wing formation, embryonic segmentation, and body axis specification [16,17,18,19]. The first mammalian *Wnt* gene (*int-1*) was identified as an uninterrupted locus activated in response to mouse mammary tumor virus insertion and was found to code for a proto-oncogene [20,21,22]. To date, there are 19 *Wnt* genes identified in mice and humans. Based on the differences in components and biological readouts, Wnt signaling pathways are divided into three branches: Wnt/β-catenin, Wnt/PCP, and Wnt/Ca^2+^ (Figure 1). Upon binding with Wnt ligands, Frizzled (Fzd) receptors and diverse proteins that function as co-receptors, including LRP5/6 (low-density lipoprotein receptor-related protein 5/6), Glypican3/4, ROR (receptor tyrosine kinase-like orphan receptor), and RYK (related to receptor tyrosine kinase), are assembled into complexes to trigger the activation of divergent downstream events [23,24,25]. In the Wnt/β-catenin branch, the activity of Fzd receptors is further modulated by auxiliary extracellular or membrane proteins including RSPO1–4 (R-Spondin1–4), RNF43/ZNRF3 (RING finger protein 43/zinc and RING finger 3), and LGR4/5/6 (leucine-rich repeat-containing G protein-coupled receptor 4/5/6) [26,27]. The activation of this pathway prevents the degradation of β-catenin by its destruction complex consisting of Axin, GSK3β (glycogen synthase kinase 3β), and APC (adenomatous polyposis coli) tumor suppressor, resulting in the stabilization of β-catenin and its translocation into the nucleus to induce target gene transcription. Therefore, this pathway plays a major role in cell fate specification [4]. The Wnt/PCP branch signals through effector proteins including Daam1 (Dishevelled associated activator of morphogenesis 1), the Rho family of small GTPases, and JNK (Jun N-terminal kinase); it modulates cytoskeletal rearrangements and/or transcriptional responses to coordinate cellular polarization [28]. The Wnt/Ca^2+^ branch triggers intracellular calcium flux and activates calcium-dependent responses through PLC (phospholipase C) and heteromeric G proteins; this signaling cascade regulates cell movements in many developmental and pathological processes [29]. It is of note that the scaffold protein Dishevelled (Dvl) is a common component of all three Wnt pathway branches and mediates the activation of downstream signals through distinct domains [30,31]. The extreme C-terminus of Dvl may be involved in modulating conformational changes in the protein to differentially activate Wnt/β-catenin and Wnt/PCP signaling [32,33,34].

It is thought that Wnt/PCP signaling regulates cellular polarization essentially through six “core” proteins. These “core” PCP components form two separate complexes which are asymmetrically distributed at cell borders within a tissue plane. Generally, Vangl complexes with Prickle to localize at the anterior/proximal side, while Fzd, Dvl, and Ankrd6 are restricted to the posterior/distal edge. The protocadherin Celsr is present in both complexes and is capable of forming homodimers between adjacent cells to propagate polarity information across cells [8,35]. This characteristic feature of “core” PCP proteins provides instructive signals to establish cell polarity for asymmetric cell movements and organ morphogenesis.

## 3. Maternal Wnt/β-Catenin Signaling Dictates Dorsal Axis Specification

The year 2024 marks the 100th anniversary of the landmark discovery in embryonic induction made by Spemann and Mangold, who found that the dorsal blastoporal lip from an early amphibian gastrula could function as an “organizer” to induce a complete secondary axis when transplanted to the ventral region of a host embryo [36]. The molecular nature of the Spemann organizer in different vertebrates is now well elucidated with the characterization of transcription and secreted factors that specify and protect this group of cells [5,7,37,38,39,40,41,42]. The *Xenopus* and zebrafish models, with their in vitro and rapid development, as well as their suitability for experimental and genetic manipulations, have significantly contributed to the understanding of Wnt signaling in embryonic axis formation. In these species, maternal Wnt/β-catenin signaling acts upstream of the Spemann organizer and critically contributes to inducing its formation (Figure 2A). As demonstrated in *Xenopus*, the accumulation of maternal β-catenin combined with a high level of Nodal signal in the dorsal–vegetal cells of the blastula constitute the Nieuwkoop center, which then induces the formation of the Spemann organizer in the overlying dorsal marginal zone [5]. Injection of synthetic mRNAs encoding canonical Wnt ligands, such as Wnt1 and Wnt8, into the ventral region of *Xenopus* early cleavage stage embryos can induce a complete secondary axis, mimicking the activity of the Spemann organizer [43,44,45]. Ventral overexpression of other components of the Wnt/β-catenin pathway in *Xenopus*, including Dvl and β-catenin, also leads to a complete axis duplication [46,47], while inhibition of GSK3β activity by lithium treatment at cleavage stages completely dorsalizes the embryo [48]. In zebrafish, the maternal-effect *ichabod* mutant embryos lack organizer formation and display an absence of dorso-anterior structures, due to reduced activity of β-catenin2 [49,50]. In *Xenopus*, the homeobox genes *Siamois* and *Twin* have been identified as direct targets of Wnt/β-catenin signaling in Spemann organizer formation [51,52]. These observations have firmly established a critical role for β-catenin in the establishment of the dorsal axis. Nevertheless, how maternal Wnt/β-catenin signaling is activated by upstream components has been a subject of debate, because it has been shown that interference with Dvl activity in the *Xenopus* embryo affects gastrulation cell movements but not dorsal axis formation [53].

With the advent of forward genetics and genome editing, maternal mutants affecting the function of Wnt pathway genes can be obtained for the analysis of their contribution to dorsal axis formation. In *Xenopus*, CRISPR-mediated maternal mutation of *Wnt11b* suggests that it is required for the dorsal distribution of maternal determinants but not for activation of Wnt/β-catenin signaling in dorsal fate specification, likely by regulating microtubule assembly and cortical rotation [54]. In zebrafish, the loss of maternal Dvl proteins does not affect the activation of Wnt/β-catenin signaling and the expression of organizer genes, suggesting that they are not required for dorsal fate specification [31,55]. Therefore, maternal Wnt/β-catenin signaling should be activated by components downstream of Dvl proteins. This is supported by the identification of a novel maternal gene in a spontaneous maternal-effect mutant line that produces ventralized phenotypes like calabashes. This gene was named *huluwa* (Chinese for “calabash children”, inspired by the Chinese TV animation series “*Calabash Brothers*” in which seven gourd brothers are endowed with special powers to defeat monsters) [56]. Huluwa (Hwa) is a previously uncharacterized protein, and it functions independently of Wnt ligands and receptors but through β-catenin to trigger the formation of the dorsal organizer in zebrafish and *Xenopus* [56]. Interestingly, Hwa protein is localized to the cell membrane on the dorsal side at blastula stages and its overexpression in the ventral region can induce a complete secondary axis [56]. Mechanistically, Hwa directly binds to tankynase and promotes its activity in degrading Axin, a negative regulator of Wnt/β-catenin signaling, thereby stabilizing β-catenin [56]. Recent studies suggest that the activity of maternal Wnt/β-catenin signaling is tightly regulated for the proper establishment of the Spemann organizer. Outside the dorsal region, particularly in the ventral side of the embryo, there are several mechanisms restricting Spemann organizer formation. The pluripotency transcription factor Nanog directly binds to TCF (T-cell factor) to prevent its interaction with β-catenin, thereby limiting β-catenin transcriptional activity [57]. The E3 ubiquitin protein ligase ZNRF3 interacts with and regulates the spatiotemporal activity of Hwa in dorsal axis formation, by promoting its lysosomal trafficking and degradation in ventral cells [58]. These observations suggest an important role of the lysosomal pathway in regulating maternal Wnt/β-catenin signaling during D-V axis formation. More recent works indicate that lysosome function is activated on the dorsal region and plays a role in sequestrating GSK3β and Axin into multivesicular bodies, thus potentiating Wnt/β-catenin signaling in the *Xenopus* early embryo [59,60]. Therefore, there are multiple mechanisms regulating the spatiotemporal activity of maternal Wnt/β-catenin in order to properly delineate the extent of the organizer field. For example, a number of other proteins, such as PTPRK (tumor suppressor protein tyrosine phosphatase receptor-type kappa), VBP1 (pVHL binding protein 1), GPX4 (glutathione peroxidase 4), and EIF4A3 (eukaryotic initiation factor 4A3), have been shown to modulate dorsal axis formation by restricting the activation of Wnt/β-catenin signaling [61,62,63,64].

## 4. Zygotic Wnt/β-Catenin Signaling in D-V and A-P Axis Patterning

There is a fascinating story on Wnt signaling in embryonic axis formation, with the treatment of sea urchin fertilized eggs using lithium solution that dates back more than 130 years [65,66]. As opposed to maternal Wnt/β-catenin signaling (Figure 2A), zygotic Wnt/β-catenin signaling functions to promote ventral and posterior development during gastrulation (Figure 2B,C). Both in *Xenopus* and zebrafish, zygotic expression of *wnt8* is restricted to the ventral and lateral regions of the early gastrula. Ectopic activation of Wnt/β-catenin signaling in the dorsal region after zygotic transcription leads to dorsal and anterior deficiencies [67]. Therefore, after organizer formation, this pathway needs to be inactivated in the dorsal region of the early gastrula to protect the dorsal cell fate and anterior development. Importantly, several extracellular Wnt antagonists, such as Frzb, Cerberus, and Dickkopf-1, are expressed in the Spemann organizer; they function to protect the Spemann organizer field and promote head development by antagonizing the ventralizing activity of Wnt8 and BMP4 (bone morphogenetic protein 4) [68,69,70,71,72]. In the presumptive neuroectoderm, Wnt/β-catenin signaling forms a gradient to specify cell fate along the A-P axis [73,74,75]. Thus, a reductionist view of Wnt/β-catenin signaling in A-P and neural patterning would have the highest activity-inducing spinal cord in the posterior region while the lowest or no activity-inducing forebrain [5,10]. However, more recent works in *Xenopus* indicate that inhibition of Wnt/β-catenin signaling does not affect spinal cord cell fates but impairs hindbrain formation. Other signals, such as BMPs and FGFs (fibroblast growth factors), may also contribute to posterior neural patterning [76]. Indeed, Wnt signaling along with BMPs and FGFs promote posterior development by counteracting anteriorly expressed signals, including retinoic acid and extracellular antagonists of Wnt and BMP signaling [5]. Moreover, studies in zebrafish show that Wnt/β-catenin signaling functions in distinct temporal phases to specify major subdivisions of the developing brain, which is dependent on dynamic changes in the transcription of target genes [77].

How the temporal changes in the transcriptional activity of Wnt/β-catenin signaling are controlled remains elusive. There is evidence that this is at least partially regulated by differential phosphorylation of the Wnt pathway effector Tcf3. It has been shown that Wnt/β-catenin signaling leads to the phosphorylation of Tcf3 by HIPK2 (the homeodomain-interacting protein kinase 2) and its dissociation from the promoter of target genes involved in posterior development, such as *Vent2* and *Cdx4* [78]. By contrast, R-spo2, which generally functions as a positive extracellular regulator of Wnt/β-catenin signaling [79], can exert an anteriorizing activity by inhibiting Tcf3 phosphorylation in a manner that is independent of Fzd receptors, RNF43/ZNRF3 and LGR4/5 [80]. These observations suggest that the activity of Tcf3 in embryonic axis patterning may be regulated in a context-dependent manner, but further investigations are needed to decipher the underlying mechanisms.

Wnt/β-catenin signaling is also required for A-P axis patterning before gastrulation in mice [12]. For example, β-catenin is necessary for the formation of the anterior visceral endoderm and the primitive streak. Mice lacking β-catenin do not form mesoderm and anterior structures, showing defects in A-P axis formation [81,82]. However, different from *Xenopus* and zebrafish, mouse β-catenin seems to regulate the A-P axis by functioning in the embryonic ectoderm [82]. In addition, the mechanism underlying Wnt/β-catenin signaling in patterning the body axis is significantly diverged between mice and other vertebrates such as *Xenopus* and zebrafish, because of differences in the mode of development, and the presence or absence of maternal β-catenin protein [83].

## 5. Wnt/PCP Pathway Regulates Morphogenetic Movements to Elongate the A-P Axis

Wnt/PCP signaling is critically required for various morphogenetic movements in all vertebrates, such as gastrulation, neurulation, and asymmetric organogenesis [8,11]. Convergence and extension (CE) movements mediated by cell intercalations are important processes that occur during gastrulation and neurulation; they contribute to elongating the A-P axis and drive tissue spreading (Figure 3). Many components of the Wnt/PCP pathway, including ligands, receptors, co-receptors, and “core” PCP proteins, regulate polarized protrusive behaviors to promote asymmetric cell movements [11].

### 5.1. Wnt Ligands

Zebrafish *wnt11f2*, also called *wnt11* and *silberblick* (*slb*), was the first Wnt ligand known to be involved in CE movements during gastrulation. Mutation of this gene disrupts cell migration and axis extension in a cell non-autonomous manner [84,85]. In *Xenopus*, a dominant negative Wnt11 mutant that specifically inhibits non-canonical Wnt signaling blocks gastrulation cell movements without affecting cell fate [86]. Recent studies indicate that Wnt11-mediated signaling is required for blastoporal lip formation and blastopore closure associated with archenteron extension [87]. Both in zebrafish and *Xenopus*, Wnt11 signaling coordinates cell shape changes and intercalation behaviors at least partially by regulating cadherin-mediated cell adhesion [88,89].

Two other non-canonical Wnt ligands, Wnt5a and Wnt5b, show both specific and redundant roles in CE movements. In zebrafish, Wnt5a regulates cell motility during gastrulation by interacting with the CD146 receptor [90], and Wnt5b induces cellular polarization through Ryk and focal adhesion kinase [91,92]. Non-canonical Wnt ligands could also trigger transcriptional responses in CE movements. It has been shown that *Xenopus* Wnt5a interacts with Ror2 and activates JNK signaling to induce the expression of PAPC (paraxial protocadherin), a transmembrane protein involved in the control of cell–cell adhesion and morphogenetic movements [93,94]. There is evidence that Wnt ligands coordinately regulate cellular polarization by providing directional cues. In the *Xenopus* gastrula, Wnt5a and Wnt11 gradient instructs the localization of the Vangl2–Prickle3 complex to the anterior borders of ectodermal cells [95]. In mice, Wnt5a and Wnt11 are required for the elongation of the A-P axis by promoting the migration of axial and paraxial mesodermal precursor cells through the regulation of epithelial–mesenchymal transition [96]. Similarly, in chick embryos, several non-canonical Wnt ligands, such as Wnt5a, Wnt5b, and Wnt11b, are expressed in the primitive streak and control the migration of axial and paraxial mesodermal cells [97,98].

### 5.2. “Core” PCP Proteins

By asymmetric localization in the cell, these proteins transduce Wnt/PCP signaling to establish cell polarity for CE movements and A-P axis elongation. Dysfunction or inappropriate regulation of “core” PCP proteins perturbs the asymmetric cellular behaviors, leading to randomization or absence of cellular protrusions [99]. In *Xenopus*, *Fzd7* is enriched in the dorsal region of the gastrula and functions in the Wnt/PCP pathway to regulate CE movements without the effect of D-V patterning [100]. Analysis of zebrafish mutants for *fzd7a* and *fzd7b* suggests a permissive role of Fzd7-mediated non-canonical Wnt signaling in regulating cell protrusion and migration of anterior axial mesendoderm [101]. There are multiple Dvl proteins in vertebrates, which often show redundant roles in various developmental processes. Dvl2 mediates Wnt/PCP signaling to control cell polarity in the dorsal mesoderm during *Xenopus* gastrulation [53,102]. Knockout of *dvl* genes in zebrafish suggests that they cooperatively regulate CE movements in a dose-dependent manner, but *dvl2* seems to play a predominant role [55]. Mice with double or triple mutations of *Dvl* genes present CE defects during neurulation and show disrupted A-P axis specification associated with impaired mesoderm differentiation; these phenotypes are also dosage sensitive and independent of Wnt/β-catenin signaling [103,104,105]. As a scaffold protein, Dvl is recruited to the plasma membrane by Fzd receptors to mediate Wnt/PCP signaling [33,106]. Celsr1 also contributes to regulating the membrane recruitment of Dvl and promoting the formation of the Fzd–Dvl complex [107]. Consistent with the asymmetric localization of “core” PCP proteins, Prickle1 is distributed in the anterior edge of cells undergoing CE movements, while Dvl shows posterior enrichment, thereby conferring distinct anterior and posterior properties and providing bias for cell intercalations [108]. However, Prickle1 may regulate gastrulation cell movements by activating both Wnt/PCP and Wnt/Ca^2+^ signaling, implying a possible overlap between these non-canonical Wnt pathways [109,110,111]. Vangl2, also known as Strabismus or Trilobite, displays dynamic accumulation at the plasma membrane to mediate mediolaterally polarized cell behavior [112]. Although both gain and loss of Vangl2 function lead to the gross phenotype of CE movements [113,114,115], a detailed analysis of cellular behaviors in zebrafish *vangl2* mutants indicates a defective convergence toward the dorsal midline and a biased anterior movement of lateral mesodermal cells [112].

In the chick embryo, mediolateral cell intercalation in a restricted ectodermal subdomain defines the primitive streak before gastrulation, a process that requires the function of several “core” PCP proteins including Dvl, Celsr1, Vangl2, and Prickle1; however, this intercalation event differs from CE movements found in *Xenopus* and zebrafish because it occurs before gastrulation and between columnar epithelial cells [116]. Disruption of the Wnt/PCP pathway prevents the proper location of mesendoderm, suggesting that Wnt/PCP signaling regulates the midline positioning of the primitive streak [116].

### 5.3. Co-Receptors

Glypican4 belongs to the family of HSPGs (heparan sulfate proteoglycans) and is localized to the plasma membrane via GPI (glycosylphosphatidylinositol) anchor; it promotes Wnt5a and Wnt11 signaling to regulate gastrulation cell movements [117,118]. Zebrafish mutants for *glypican4*, previously known as *knypek* (*kny*), show CE defects and a shortened A-P axis due to disrupted cell polarity and defective mediolateral alignment that prevent elongation of ectodermal and mesodermal cells in the paraxial region [101]. Ror2 and Ryk also enhance Wnt5a and Wnt11 signaling to regulate CE movements by interacting with Fzd7 receptor [119,120,121,122].

Zebrafish maternal–zygotic mutants for *ptk7* (protein tyrosine kinase 7) show impaired CE of axial tissues, and knockout of *Ptk7* in mice impairs gastrulation movements due to defective mediolateral and radial intercalations [123,124]. Intriguingly, CE defects in zebrafish *ptk7* mutants can be rescued by a membrane-tethered extracellular domain of the protein [123]. Thus, how Ptk7 activates Wnt/PCP signaling needs further investigation. In *Ptk7* mutant mouse embryos, cells fail to undergo elongation and alignment upon leaving the primitive streak, which subsequently leads to defective polarized protrusive activity, abnormal CE movements, and impaired axial extension [124]. CE movements of the neural plate drive axial elongation in mammalian embryos. The loss of *Ptk7* in mice also impairs mediolateral intercalation and causes defects in the neural tube [125,126]. In addition, *Ptk7* shows genetic interaction with *Vangl2* in neural tube closure [125]. Importantly, missense variants in *PTK7* are associated with neural tube defects in humans [127].

Overall, different components of the Wnt/PCP pathway play a critical role in coordinating cell movements during gastrulation, which is important for elongating and positioning the A-P axis. It is of note that disrupted Wnt/PCP signaling prevents the extension of axial tissues and affects the positioning of the eye primordium, leading to cyclopia, neural tube defects, and craniofacial malformations [11]. These phenotypes are frequently present in zebrafish mutants with loss of PCP genes, such as *silberblick*/*wnt11*, *vangl2*, *knypek*/*glypican4*, and *dvl* [55,74,118,128,129,130]. Therefore, mutations of PCP genes can contribute to a broad spectrum of birth defects.

## 6. Wnt/PCP Signaling Initiates L–R Asymmetry

### 6.1. L–R Organizers

The establishment of L–R asymmetry, either external or internal, is a fundamental process in development, which dictates the asymmetric location of internal organ primordia, such as the heart and liver. Vertebrate embryos initially display bilateral symmetry, but ciliated transient organs formed during early development establish gene expression differences across the mediolateral plane (Figure 4). These transient structures constitute the L–R organizer, including Kupffer’s vesicle (KV) in the zebrafish early segmentation stage embryo, the posterior gastrocoel roof plate in the *Xenopus* early neurula, the Hensen’s node in chicks, and the node in mice [8]. It is thought that at least in zebrafish, *Xenopus*, and mice, the L–R organizer breaks the bilateral symmetry by providing mechanosensory or chemosensory signals through cilia-driven directional fluid flow [131,132,133]. There is evidence that Wnt/β-catenin signaling functions to specify cell fate during L–R formation. This aspect will not be further discussed here because it has been recently reviewed in detail elsewhere [134]. Wnt/PCP signaling, however, acts to coordinate the orientation of motile cilia within the L–R organizer, thus initiating the early asymmetry development. Subsequently, a leftward fluid flow generated by the clockwise rotational motion of motile cilia within the cavity of the L–R organizer, known as Nodal flow, contributes to creating a gradient of Nodal protein across the L–R axis and activates the left-sided expression of the Nodal–Lefty–Pitx2 network [131,135]. This differential gene expression will influence asymmetric organ morphogenesis [136,137,138].

### 6.2. Wnt/PCP Signaling Promotes the Asymmetric Orientation of Motile Cilia

Wnt ligands are important for initiating the cellular asymmetry in the L–R organizer. In *Xenopus*, Wnt11b-dependent Wnt/PCP signaling is required for the polarization of cilia in the gastrocoel roof plate [119]. The loss of Wnt11b disrupts leftward fluid flow and asymmetric gene expression, leading to heterotaxy and abnormal gut coiling [54,139]. In the mouse embryo, Wnt5a and Wnt5b are expressed posteriorly relative to the node; they form a diffusible gradient that initiates the asymmetric localization of “core” PCP proteins in node cells [140]. As a result, Dvl2 and Dvl3 are enriched at the posterior cell borders, while Vangl1, Vangl2, and Prickle2 accumulate at the anterior side; Celsr1 is present at both anterior and posterior sides [135,141,142]. The asymmetric localization of “core” PCP proteins leads to a biased distribution of microtubules and actomyosin networks, which contribute to positioning the ciliary basal bodies at the posterior side of node cells and restricting the posterior tilting of cilia [143]. The dysfunction of Wnt/PCP signaling mediated by “core” PCP proteins prevents L–R asymmetry development by disrupting the positioning of cilia and the left-sided expression of the *Nodal* gene. Although ciliary basal bodies show the correct location in mice lacking any of the three *Dvl* genes, they fail to shift posteriorly after the deletion of five *Dvl* alleles (with only one *Dvl3* allele), suggesting that Dvl proteins play redundant roles in cilia orientation [143]. Similarly, the loss of Vangl1 and Vangl2 also affects the posterior orientation of motile cilia in different species, including zebrafish [144], *Xenopus* [141], and mice [142,145,146]. The asymmetric distribution of “core” PCP proteins is also dependent on their interactions. Prickle1 and Prickle2 regulate the anterior localization of Vangl1 to promote the A-P polarization of node cells [140]. In *Xenopus*, Prickle3 and Vangl2 show interdependent localization at the anterior borders of gastrocoel roof plate cells, which promotes cilia growth and posterior positioning [147]. Altogether, these observations suggest that the A-P polarity of the L–R organizer established by “core” PCP proteins is translated into L–R asymmetry through cilia-driven directional fluid flow and subsequent expression of laterality genes [148].

The molecular events initiated by cilia-driven directional fluid flow are partially understood. Dand5, previously known as Cer2 or Cerl2, is an extracellular antagonist of Nodal protein and functions to prevent Nodal signaling [149]. It is the first gene asymmetrically expressed in the L–R organizer and involved in L–R patterning. There is evidence that *Dand5* mRNA is subjected to selective degradation on the left side of the mouse node, resulting in its expression only on the right side [150]. Mechanistically, the RNA-binding protein Bicc1 (Bicaudal C) promotes the degradation of *Dand5* mRNA at the left side by binding to its 3′-untranslated region [151,152]. As a consequence, this increases Nodal signaling and induces the expression of Nodal, Lefty, and Pitx2 on the left side of the lateral plate mesoderm. Thus, Dand5 functions downstream of Wnt/PCP signaling and represents an early flow target gene in L–R patterning. Studies in mice suggest that Wnt/β-catenin signaling regulates the asymmetric expression of Dand5 [153], and that the leftward flow can be enhanced by Wnt–Dand5 interlinked feedback loops [154]. Wnt3 shows L–R differences in expression and promotes *Dand5* mRNA decay, while Dand5 also induces Wnt3 degradation [154]. Therefore, it will be of interest to understand how Wnt/β-catenin signaling interacts with the non-canonical Wnt pathway and post-transcriptional regulatory factors to orchestrate the left–right differences in gene expression.

### 6.3. Laterality Defects Associated with Dysfunction of PCP Genes

Since proper L–R patterning is important for asymmetric organ development, defective morphogenesis of the L–R organizer leads to laterality defects [9,134]. Recent studies have identified missense mutations in human *VANGL2* associated with heterotaxy and congenital heart disease [155]. Although the Zic3 transcription factor is only expressed in the L–R organizer but not in the heart primordium, there is evidence that it regulates the expression of PCP genes and is required for L–R asymmetry development [156,157]. In humans, mutations of the *ZIC3* gene cause X-linked situs abnormalities ranging from partially inverted to completely reversed positioning of internal organs [158]. Therefore, dysfunction of PCP genes can severely affect asymmetric organogenesis, but further studies are necessary to determine how mutations of other PCP genes in humans affect the establishment and L–R asymmetry and the development of laterality.

## 7. Conclusions and Perspectives

Wnt signaling plays a pivotal role in initiating embryonic polarity, which is evolutionarily conserved despite critical differences in the temporal and spatial activation of the pathway. Our understanding of the eminent implication of Wnt signaling in establishing the basic body plan is rapidly evolving. The challenge remains to decipher the regulation of the canonical and non-canonical Wnt pathways in key developmental processes. Wnt/β-catenin and Wnt/PCP signaling initiate the molecular asymmetry in the early embryo and are critical for the specification and subsequent development of all three embryonic axes. Maternal Wnt/β-catenin signaling establishes the D-V asymmetry and induces the formation of the Spemann organizer, which not only further patterns the D-V axis but also promotes A-P axis development during gastrulation. By contrast, zygotic Wnt/β-catenin signaling mostly contributes to posterior development and L–R organizer formation. Therefore, the spatial and temporal regulation of Wnt/β-catenin signaling is crucial for the proper formation of embryonic axes. Indeed, the identification of novel maternal components of this pathway, such as Hwa [56], and tissue-specific processes restricting its activation, such as lysosomal trafficking [58,59,60], greatly contributes to deciphering molecular mechanisms underlying embryonic axis specification and patterning. Wnt/PCP signaling functions as a key regulator of cell polarity and is essential for asymmetric morphogenesis. It orchestrates gastrulation cell movements to elongate the A-P axis and coordinates cellular orientation in the L–R organizer to break the bilateral symmetry. Obviously, there exists a close interconnection of Wnt/β-catenin and Wnt/PCP signaling in the specification of cell fate and the establishment of cell polarity. The integration of different processes regulated by both pathways sets up the basic body plan in vertebrates. Moreover, the interaction of Wnt signaling with other key developmental signaling pathways is critical for setting up the three embryonic axes. Dysfunction of Wnt pathway components not only impairs axis formation but also causes inherited disorders, as exemplified by neural tube defects and laterality defects caused by mutations of *PTK7* and *VANGL2* genes [127,156]. Therefore, a better understanding of the genetic cascade involved in embryonic axis formation will contribute to deciphering the mechanism underlying asymmetric organ morphogenesis.

## Figures and Tables

**Figure 1 jdb-12-00020-f001:**
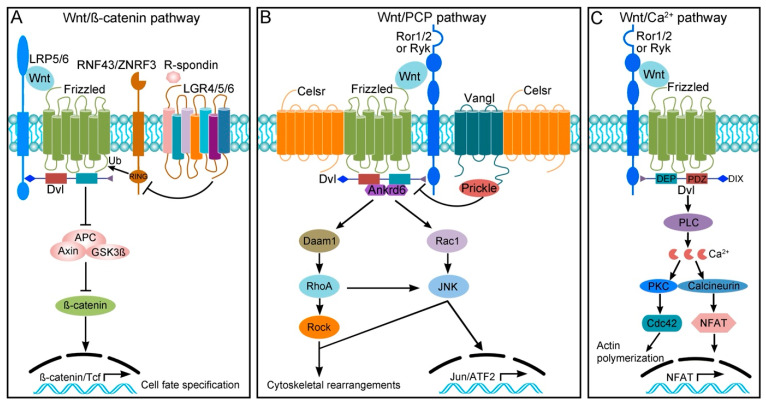
Wnt signaling pathways in vertebrates. (**A**) In the Wnt/β-catenin pathway, the binding of ligands to Fzd receptors and LRP5/6 co-receptors leads to the stabilization of β-catenin and transcription of target genes. The E3 ubiquitin ligases RNF43 and ZNRF3 function to regulate the lysosomal degradation of Fzd receptors by promoting their ubiquitination (Ub). This activity is antagonized by the binding of R-spondins to LGR4/5/6. (**B**) Wnt/PCP signaling is induced and propagated through the interaction between non-canonical Wnts and receptor–co-receptor complexes (Fzd/Ror1/2 or Fzd/Ryk) as well as by the asymmetric localization of “core” PCP proteins. The signal is relayed by downstream effectors which regulate cytoskeletal rearrangements or activate transcriptional responses. (**C**) The Wnt/Ca^2+^ branch activates PLC through heteromeric G proteins to trigger calcium-dependent cytoskeletal changes and NFAT-mediated target gene transcription. Dvl proteins contribute to activating different Wnt pathways through distinct domains: N-terminal DIX, central PDZ, C-terminal DEP, and extreme C-terminus. It should be noted that although Ryk and Ror are often associated with controlling polarized cell behaviors, they may be also involved in modulating canonical Wnt signaling [23].

**Figure 2 jdb-12-00020-f002:**
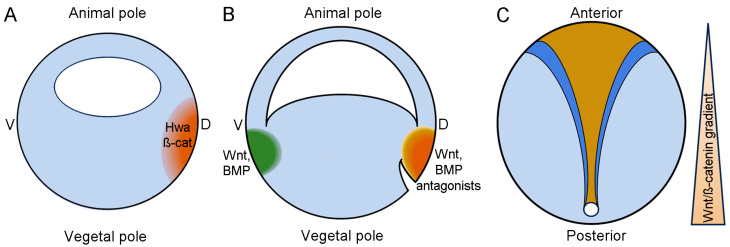
Simplified model of Wnt/β-catenin signaling in the specification of D-V and A-P axes during *Xenopus* development. (**A**) At cleavage stages, maternal Hwa and β-catenin (β-cat) are accumulated in the dorsal–vegetal region as a result of cortical rotation and selective protection. The dorsal-vegetal blastomeres with high levels of β-catenin and Nodal proteins constitute the Nieuwkoop center. Activation of maternal Wnt/β-catenin signaling will induce the formation of the Spemann organizer after zygotic transcription. (**B**) In the gastrula, the Spemann organizer region secretes extracellular inhibitors for Wnts and BMPs to prevent their ventralizing activity. This antagonistic interaction patterns the D-V axis. (**C**) During and after gastrulation, Wnt/β-catenin signaling is involved in A-P patterning, with higher activity at the posterior region of the embryo.

**Figure 3 jdb-12-00020-f003:**
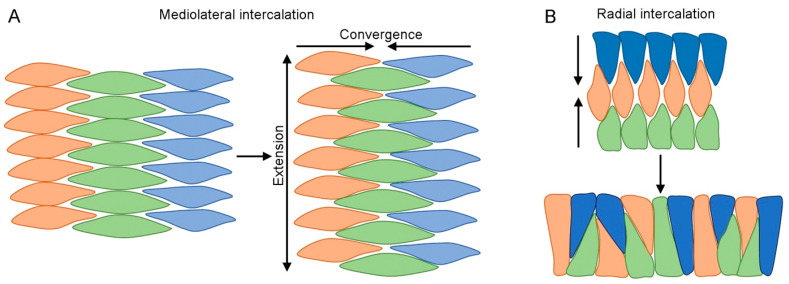
Schematic of asymmetric cellular behaviors regulated by Wnt/PCP signaling in A-P axis elongation. (**A**) Mediolateral cell intercalation in CE movements during gastrulation narrows tissues along the mediolateral plane and elongates the embryo along the A-P axis. (**B**) Radial cell intercalation reduces the number of cell layers and drives tissue spreading.

**Figure 4 jdb-12-00020-f004:**
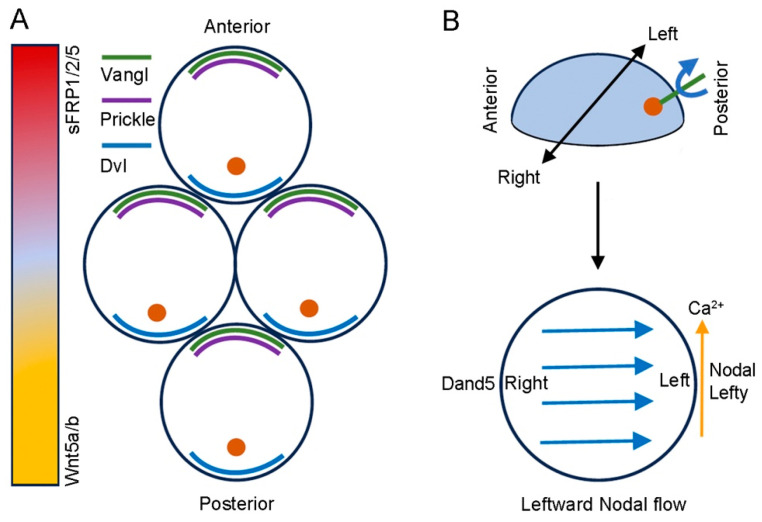
Schematic of “core” PCP protein localization and L–R asymmetry formation in the mouse node. (**A**) The posterior expression of Wnt5a and Wnt5b in the node forms a gradient of Wnt/PCP signaling along the A-P axis to initiate the asymmetric localization of “core” PCP proteins. In the anterior region of the node, high levels of Wnt antagonists sFRP1/2/5 prevent Wnt/PCP signaling. The asymmetric localization of “core” PCP proteins contributes to restricting the posterior positioning of ciliary basal bodies (orange dots) in node cells. (**B**) At the dome-shaped apical surfaces of node cells, the posterior tilting and the clockwise rotational motion of motile cilia generate leftward fluid flow (blue arrows) within the node cavity, resulting in an increased calcium concentration on the left side (vertical yellow arrow). This Nodal flow triggers left-sided gene expression and breaks the bilateral symmetry.

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
