# Peer review of "Canonical and Non-Canonical Wnt Signaling Generates Molecular and Cellular Asymmetries to Establish Embryonic Axes"

_jdb, 2024, doi:10.3390/jdb12030020_

Round 1

Reviewer 1 Report

Comments and Suggestions for Authors

The presented manuscript addresses a fundamental problem of embryo patterning via Wnt signaling pathway. The topic of the work is extremely relevant in connection with the unresolved problems of understanding the molecular mechanisms of development and the causes of congenital developmental anomalies. The author reveals in detail the biochemical and genetic details of the implementation of various branches of canonical and non-canonical Wnt signaling. The demarcating line between the different modes of action of the autonomous maternal and inductive zygotic Wnt/beta-catenin pathways is very well drawn. The strength of the work must be recognized as the extremely correct and modern display of data from genetics and developmental biology, which provides a comprehensive overview of morphogenetic processes in the specification of body axes.

The manuscript is very well and logically written; the illustrations are of very good quality.

General comments

Although the author outlined the focus of his review on vertebrates, it is extremely important to give an evolutionary aspect of the participation of Wnt in the control of embryonic axes. Particularly valuable here are works on non-bilateral organisms, in which the role of Wnt in the development of animal-vegetative polarity has been revealed. It is necessary to mention at least the latest review articles:

Holstein, T.W., 2022. The role of cnidarian developmental biology in unraveling axis formation and Wnt signaling. Developmental Biology 487, 74–98.

Kozin, V.V., Borisenko, I.E., Kostyuchenko, R.P., 2019. Establishment of the Axial Polarity and Cell Fate in Metazoa via Canonical Wnt Signaling: New Insights from Sponges and Annelids. Biol Bull Russ Acad Sci 46, 14–25.

Specific comments

1) line 116: decipher Hwa the first time it is mentioned.

2) line 135: complete secondary (axis?)

3) Fig. 4B: Explain what Ca2+ means on the left side of the L-R organizer.

4) line 332: the phrase “morphogenesis of ... cells” does not look very good.

Author Response

The presented manuscript addresses a fundamental problem of embryo patterning via Wnt signaling pathway. The topic of the work is extremely relevant in connection with the unresolved problems of understanding the molecular mechanisms of development and the causes of congenital developmental anomalies. The author reveals in detail the biochemical and genetic details of the implementation of various branches of canonical and non-canonical Wnt signaling. The demarcating line between the different modes of action of the autonomous maternal and inductive zygotic Wnt/beta-catenin pathways is very well drawn. The strength of the work must be recognized as the extremely correct and modern display of data from genetics and developmental biology, which provides a comprehensive overview of morphogenetic processes in the specification of body axes.

The manuscript is very well and logically written; the illustrations are of very good quality.

Author: Thank you for your enthusiastic assessment of this work.

General comments

Although the author outlined the focus of his review on vertebrates, it is extremely important to give an evolutionary aspect of the participation of Wnt in the control of embryonic axes. Particularly valuable here are works on non-bilateral organisms, in which the role of Wnt in the development of animal-vegetative polarity has been revealed. It is necessary to mention at least the latest review articles:

Holstein, T.W., 2022. The role of cnidarian developmental biology in unraveling axis formation and Wnt signaling. Developmental Biology 487, 74–98.

Kozin, V.V., Borisenko, I.E., Kostyuchenko, R.P., 2019. Establishment of the Axial Polarity and Cell Fate in Metazoa via Canonical Wnt Signaling: New Insights from Sponges and Annelids. Biol Bull Russ Acad Sci 46, 14–25.

Author: Thank you for the insightful comments and helpful suggestions. The conservation of Wnt signaling in primary body axis formation across the Metazoa is mentioned and the references above are cited in the revised manuscript.

Specific comments

1) line 116: decipher Hwa the first time it is mentioned.

Author: Thank you for raising this point. It is now explained in the revised manuscript.

2) line 135: complete secondary (axis?)

Author: Thanks for careful examination. Indeed, it is “complete secondary axis”.

3) Fig. 4B: Explain what Ca2+ means on the left side of the L-R organizer.

Author: It is now explained in the legend.

4) line 332: the phrase “morphogenesis of ... cells” does not look very good.

Author: This sentence is changed to “Wnt ligands are important for initiating the cellular asymmetry in the L-R organizer”.

Reviewer 2 Report

Comments and Suggestions for Authors

The review "Canonical and non-canonical Wnt signaling generates molecular and cellular asymmetries to establish embryonic axe" by De-Li Shi offers at once a comprehensive survey and a very narrow point of view. This comes from the way the abstract mentions the big picture of the Wnt field, but then the text focuses mainly on zebrafish and xenopus findings, with even the vertebrate, mouse, being relatively underrepresented. I know that this is a big field, but at least the introduction and conclusion should be revised to show the bigger picture.

The historical points were very good, with Spemann's 100th anniversary mentioned, and the original Lithium experiments mentioned. This should cite the original papers rather than the recent review (Ref. 54 contains the original citations). But raising these historical points also highlights the lack of other highlights such as the discovery of wingless and the cloning of Int-1. 

Overall, I think the abstract needs to reflect the content more directly rather than giving a hasty overview of the field. That should be in the introduction.

The figures are informative. Figure 1 does seem to suggest that Ryk/Ror-1/Ror-2 are the non-canonical receptors in all cases. I don't believe that this has been definitively proven, so a caveat should be added. This is particularly relevant as there is a paragraph on PTK7. 

The PTK7 paragraph should be either expanded or removed. There is a lot more known about PTK7, especially in mice and flies than is mentioned and cited.

Comments on the Quality of English Language

English is mostly fine. Minor typos.

Author Response

The review "Canonical and non-canonical Wnt signaling generates molecular and cellular asymmetries to establish embryonic axe" by De-Li Shi offers at once a comprehensive survey and a very narrow point of view. This comes from the way the abstract mentions the big picture of the Wnt field, but then the text focuses mainly on zebrafish and xenopus findings, with even the vertebrate, mouse, being relatively underrepresented. I know that this is a big field, but at least the introduction and conclusion should be revised to show the bigger picture.

Author: Thank you for the critical and insightful comments. The abstract, introduction and conclusion sections are modified to reflect the content directly. The manuscript is also modified by including more studies in mice.

The historical points were very good, with Spemann's 100th anniversary mentioned, and the original Lithium experiments mentioned. This should cite the original papers rather than the recent review (Ref. 54 contains the original citations). But raising these historical points also highlights the lack of other highlights such as the discovery of wingless and the cloning of Int-1. 

Author: Thank you for raising these issues. The original paper reporting lithium experiments is cited. The discovery of Wingless and the cloning of int-1 are also highlighted in the revised manuscript by citing several original papers (Sharma and Chopra, 1976; Nüsslein-Volhard and Wieschaus, 1980; Nusse and Varmus, 1982; Nusse et al., 1984; Cabrera et al., 1987; Rijsewijk et al., 1987; McMahon and Moon, 1989).

Overall, I think the abstract needs to reflect the content more directly rather than giving a hasty overview of the field. That should be in the introduction.

Author: Thank you for this recommendation. The abstract is modified to reflect the content.

The figures are informative. Figure 1 does seem to suggest that Ryk/Ror-1/Ror-2 are the non-canonical receptors in all cases. I don't believe that this has been definitively proven, so a caveat should be added. This is particularly relevant as there is a paragraph on PTK7. 

Author: Thank you for raising this point. This is mentioned in legend.

The PTK7 paragraph should be either expanded or removed. There is a lot more known about PTK7, especially in mice and flies than is mentioned and cited.

Author: This paragraph is expanded by discussing and citing works in mice and mentioning the association of missense variants of PTK7 with neural tube defects in humans.

Reviewer 3 Report

Comments and Suggestions for Authors

This manuscript provides a comprehensive review of Wnt pathways' role in establishing early embryo asymmetries essential for developing the dorsoventral, anteroposterior, and left-right axes. The review discusses recent progress in understanding these mechanisms and the challenges in deciphering Wnt pathway regulation in development.

The manuscript is well-written and provides a nice summary of the role of canonical and non-canonical Wnt signaling in body axis formation. One minor aspect is that it mainly summarizes studies from anamniotes like Xenopus and zebrafish, with limited inclusion of studies from amniotes. Moreover, it is also important to discuss interactions between Wnt signaling and other signaling pathways crucial for setting up the three body axes. Also, it will be beneficial to mention the Nieuwkoop center, which provides localized Wnt signaling to induce the Spemann organizer.

Author Response

This manuscript provides a comprehensive review of Wnt pathways' role in establishing early embryo asymmetries essential for developing the dorsoventral, anteroposterior, and left-right axes. The review discusses recent progress in understanding these mechanisms and the challenges in deciphering Wnt pathway regulation in development.

The manuscript is well-written and provides a nice summary of the role of canonical and non-canonical Wnt signaling in body axis formation. One minor aspect is that it mainly summarizes studies from anamniotes like Xenopus and zebrafish, with limited inclusion of studies from amniotes. Moreover, it is also important to discuss interactions between Wnt signaling and other signaling pathways crucial for setting up the three body axes. Also, it will be beneficial to mention the Nieuwkoop center, which provides localized Wnt signaling to induce the Spemann organizer.

Author: Thank you for your positive assessment of this work and for raising insightful comments and suggestions. In the revised manuscript, more studies from mice have been included in the discussion. For example, the role of ß-catenin in the formation of the anterior visceral endoderm and the primitive streak, the role of Dvl genes in A-P specification, and the role of Wnt/PCP signaling in the morphogenesis of the L-R organizer. The interactions between Wnt signaling and other developmental signaling pathways are also mentioned in some places, such as the cooperation between ß-catebin and Nodal in the formation of the Nieuwkoop center, the interactions between Wnt signaling and FGF and BMP signaling in A-P patterning, the mutual regulation of Wnt and Dand5 in left-right asymmetry. Finally, the formation of the Nieuwkoop center and its role in inducing the Spemann organizer is mentioned in the section “Maternal Wnt/ß-catenin signaling dictates dorsal axis specification”.

Round 2

Reviewer 1 Report

Comments and Suggestions for Authors

The author has taken into account all the comments of the reviewer, now the manuscript can be accepted for publication.

Reviewer 2 Report

Comments and Suggestions for Authors

Thank you for addressing the comments.

Reviewer 3 Report

Comments and Suggestions for Authors

The authors have addressed all my previous comments in the revised manuscript.